# Cup Overhanging in Anatomic Socket Position or High Hip Center of Rotation in Total Hip Arthroplasty for Crowe III and IV Dysplasia: A CT-Based Simulation

**DOI:** 10.3390/jcm12020606

**Published:** 2023-01-12

**Authors:** Francesco Castagnini, Federico Giardina, Chiara Fustini, Enrico Tassinari, Barbara Bordini, Monica Cosentino, Francesco Traina

**Affiliations:** 1Ortopedia-Traumatologia e Chirurgia Protesica e dei Reimpianti d’anca e di Ginocchio, IRCCS Istituto Ortopedico Rizzoli, Via Pupilli 1, 40136 Bologna, Italy; 2Laboratorio di Tecnologia Medica, IRCCS Istituto Ortopedico Rizzoli, Via Pupilli 1, 40136 Bologna, Italy; 3DIBINEM, University of Bologna, Via Irnerio 48, 40126 Bologna, Italy

**Keywords:** cup protrusion, iliopsoas impingement, anterior soft tissue irritation, groin pain, tendonitis

## Abstract

Cup overhanging in total hip arthroplasty is a predisposing factor to iliopsoas impingement. In dysplastic hips, cup implantation was simulated in an anatomic hip center of rotation (AHCR) and in high hip center (HHCR). We sought to assess: (1) the percentage of prominent cups; (2) quantify the cup protrusion at different sites on frontal, axial and sagittal views. In 40 Crowe III-IV hips, using a 3D CT-based planning software, cup planning in AHCR and HHCR (CR height ≥ 20 mm) was performed for every hip. Cup prominence was assessed on every plane. HHCR cups were less anteverted (*p* < 0.01), less medialized (*p* < 0.001) and less caudal (*p* = 0.01) than AHCR sockets. AHCR cups were more frequently prominent on at least one plane (92.5% vs. 77.5%), with minimal agreement between the two configurations (k = 0.31, *p* = 0.07). AHCR cups protruded more than HHCR sockets in the sagittal (*p* = 0.02) and axial planes (*p* < 0.001). Axially, at the center of the cup, prominence 6–11 mm occurred in nine (22.5%) AHCR and one (2.5%) HHCR socket. In conclusion, while a routine high hip center should not be recommended, cup placement at a center of rotation height < 20 mm is associated with higher rates and magnitudes of anterior cup protrusion in severe dysplasia.

## 1. Introduction

Iliopsoas impingement is an uncommon cause of groin pain after total hip arthroplasty (THA), occurring in 2–5% of the implants [1,2]. Cup overhanging has been advocated as one of the most important causative factors: Dora et al. and Cyteval et al. reported that an anterior cup prominence between 5.8 and 12 mm is necessary, but not sufficient, to cause iliopsoas impingement on the acetabular component [3,4]. Cup prominence may be due to cup oversizing or version mismatch between the acetabular component and the native acetabulum; THA for developmental dysplasia of the hip (DDH) may be more prone to iliopsoas impingement, due to the anterior and superolateral bony deficiency and the small sized acetabulum [5]. There is a paucity of literature providing data about the incidence of iliopsoas impingement and possible causative factors in THA for DDH. Only a comparative study by Zhu et al. reported that iliopsoas impingement occurred in 2.6% of the THAs performed in highly dysplastic hips with no prior surgical release, whereas the incidence in non-dysplastic hips dropped to 0.8% [5]. Moreover, there is no study investigating the incidence of cup overhanging in DDH when the cup was positioned reproducing the anatomic hip center of rotation (AHCR) or when a high hip center of rotation (HHCR) was adopted. A recent systematic review about comparative studies investigating the center or rotation height found out that, although HHCR had higher rates of dislocations and lower rates of neurological injuries, both the configurations achieved similar clinical and radiographic outcomes, even at long-term, and no recommendation about center of rotation height was formulated [6]. However, iliopsoas tendinopathy and cup overhanging were not mentioned [6].

Thus, we selected a series of Crowe III and IV dysplastic hips. Using a 3D CT-based pre-operative planning software, a cup implantation was simulated in AHCR and HHCR, in every hip. We sought to: (1) determine the percentage of prominent cups in high-riding dysplasia in AHCR and HHCR, (2) quantify the cup protrusion at different sites on frontal, axial and sagittal views in AHCR and HHCR. 

## 2. Materials and Methods

The institutional review board approved the study (IRB 29/2021/Oss/IOR, 1 February 2021).

From the hospital CT database, collecting scans from 2000, a random series of 45 pelvis CTs with native dysplastic hips graded III or IV according to Crowe was selected [5]. Other forms of congenital pathologies or lower degrees of dysplasia, as well as non-native hips, were excluded.

The demographic features of the selected patients were collected.

Every CT scan was evaluated using a 3D CT-based pre-operative planning software, Hip-Op [7]. The software is a CT-based, 3D planning environment with a user-friendly graphic user interface, based on a multimodal display visualization. The implants and the patient anatomy are rendered in each view. The planner may select a prosthetic component from a library, and may evaluate the appropriate implant type, size and position by interactively moving and rotating the components in the view area. Pre- and post-planning measurements can be performed using pre-determined and customized techniques. 

The anatomy of every native hip was evaluated by collecting data about the center of rotation height, neck-shaft angle, femoral offset, acetabular offset and acetabular anteversion. All the measurements were performed by the first author, using techniques previously detailed in other papers [7]. Inter/intra-observer reliability was proved in another paper [7].

In every hip, the first author simulated the implantation of a standard-sized acetabular cup (Continuum Zimmer, Warsaw, US) both in AHCR and HHCR position. In AHCR, the acetabular component was placed in the true acetabulum with the aim to place the center of rotation in the anatomic position; the anatomic position was cross-checked by assessing the center of rotation height, which should be <20 mm [6]. Differently, in HHCR, the cup center of rotation was positioned outside the true acetabulum (and the anatomic center of rotation of the hip), where the superolateral bony coverage was most adequate: a center of rotation height ≥ 20 mm from the inter-teardrop line (vertical distance) was required [6]. The 20 mm threshold was obtained from the vertical distance ranges provided in the comparative study by Nawabi et al.; it was a low threshold for the center of rotation height, in line with many dedicated comparative studies about AHCR and HHCR (and around 15 mm lower than the classical high hip center proposed by Harris et al.) [6]. For both the solutions, the cup abduction and the cup anteversion targets were set at 40–45° and 10–20°, respectively [7]. Some adjustments were allowed to maximize the cup coverage; however, a cup inclination higher than 50° and a cup retroversion or a cup anteversion higher than 30° were not admitted. A superior cup undercoverage of more than 33% of the socket surface or pelvic disjunction due to anterior or posterior column violation were not admitted as well. If the cup could not be placed in both AHCR and HHCR positions, the case was excluded.

The planner detailed the cup size, the cup positioning (anteversion, abduction, medialization, center of rotation height, distance between the anterior margin of the native acetabulum and the most anterior surface of the cup on the axial view at the center of the cup), in AHCR and HHCR [7,8,9].

In AHCR and HHCR, the outcome measurements for all the included cases were measured on the three planes, at 5 different sites, as follows: superior cup overhanging on the frontal view (CT slice passing at the center of the cup); anterior cup overhanging on the sagittal view (CT slice passing at the center of the cup); anterior cup overhanging on the axial view (CT slides passing at the superior edge of the cup, in the middle of the superior half of the cup and at the center of the cup). The correct segmentation was provided by the software [Figure 1].

The percentages of prominent cups on every plane, on every site were determined, in AHCR and HHCR. The percentages of cups with a prominence 6–11 mm and over 12 mm on every plane and on every site were assessed: the target values were provided in literature [3,4].

All the measurements were taken as distances, considering the most prominent bony landmarks and the most prominent part of the cup on each view.

The inter/intra-rater reliability of the first author as planner was assessed in a previous paper [7].

### Statistical Analysis

The analysis was performed using SPSS 14.0 (SPSS Inc., Chicago, IL, USA). Quantitative data were reported as average values, standard deviations and ranges of minimum and maximum; qualitative data were expressed as frequencies and percentages. The ordinal variables of the two cohorts were compared using the non-parametric Wilcoxon test. Cohen’s kappa coefficient (κ) was used to measure the agreement of the two configurations at the 5 sites; the *p*-value for kappa was calculated to determine whether to reject or not the following null hypotheses. The Wilcoxon signed test rank was adopted to test the difference between the two configurations. The threshold for significance was set at *p* = 0.05.

## 3. Results

### 3.1. Demographics and Native Hip Features

45 Crowe III and IV dysplastic hips in 40 patients were selected; in five cases, the simulation could not be performed both in AHCR and HHCR configurations and were excluded. The final cohort included 40 hips in 35 patients. The cohort encompassed 9 males (25.7%) and 26 females (74.3%), with the following demographics: mean age at CT 57.1 ± 10.9 years (range: 27–73), mean height 154.3 ± 10.1 cm (range: 130–175), mean weight 64.8 ± 17.3 kg (range: 42–112), mean BMI 27.1 ± 6.1 kg/m^2^ (range: 17.5–38.8). The native hips were classified as Crowe III (25, 62.5%) or Crowe IV (15, 37.5%) and showed the following anatomic features: mean center of rotation height 36.6 ± 12.6 cm (range: 20–77), mean neck-shaft angle 134.6 ± 18° (range: 97–160), mean acetabular anteversion 17.8 ± 10.3° (range: 0–37), mean acetabular offset 38 ± 6.6° (range: 28–55).

### 3.2. Cup Planning in AHCR and HHCR

In AHCR configuration, 37 simulations (92.5%) adopted a 44 mm cup and 3 simulations adopted a 46 mm cup (7.5%), whereas in HHCR, all the cups were 44 mm (100%) (*p* = 0.08). The mean cup anteversion was 19.7 ± 5.1° (range: 10–27) in AHCR and 14.5 ± 7.8° (range: 5–26) in HHCR (*p* < 0.01). The mean cup abduction was 43.6 ± 1.6° (range: 38–45) in AHCR and 43.8 ± 1° (range: 41–45) in HHCR (*p* = 0.79). The mean cup medialization was 27 ± 1.5 mm (range: 22–29) in AHCR and 31.2 ± 2.8 mm (range: 25–37) in HHCR (*p* < 0.001). The mean center of rotation was 16.7 ± 2.1 mm (range: 11–19) in AHCR and 29.4 ± 6.4 mm (range: 22–43) in HHCR (*p* < 0.001). The distance between the anterior margin of the native acetabulum and the most anterior surface of the cup on the axial view at the center of the cup was 8.4 ± 0.6 mm (range: 2–26) in AHCR and 6.3 ± 3.7 mm (range: 2–19) in HHCR (*p* = 0.01).

### 3.3. Percentage of Cup Overhanging in AHCR and HHCR

Prominent cups on at least a single site were 37 (92.5%) in AHCR and 28 (77.5%) in HHCR. A cup protrusion could be detected on the three different planes, at five different sites, in six (15%) AHCR cases and two (5%) HHCR cases. Nine cups (22.5%) overhung in AHCR, but not in HHCR, and three cups (7.5%) were not prominent regardless the configuration. Considering every site, there was a minimal agreement in terms of cup overhanging between the two configurations (k = 0.31, *p* = 0.07).

### 3.4. Cup Overhanging at Different Sites in AHCR and HHCR

Superior cup overhanging on the frontal view (at the center of the cup) was not evident in 13 (32.5%) AHCR cups and 14 (35%) HHCR sockets. Overhanging between 6–11 mm was evident in 17 (42.5%) AHCR and HHCR cups, and 5 (12.5%) AHCR and 4 (10%) HHCR sockets protruded more than 12 mm [Table 1]. 

No agreement between the two configurations could be detected (k = 0.03, *p* = 0.78) and no significant difference between the two configurations could be observed (*p* = 0.45). 

Anterior cup overhanging on the sagittal view (at the center of the cup) was not detected in 18 (45%) AHCR cups and 32 HHCR cups (80%). Nine (22.5%) AHCR and five (12.5%) HHCR cups overhung between 6–11 mm, and only one (2.5%) AHCR cup protruded more than 12 mm. No agreement between the two configurations could be detected (k = 0.14, *p* = 0.36) and a significant difference between the two configurations could be noted (*p* = 0.02). 

No anterior cup overhanging on the axial view at the superior edge of the cup was detected in 23 (57.5%) AHCR and 31 (77.5%) HHCR sockets. The cup protruded 6–11 mm in twelve (30%) AHCR and one (2.5%) HHCR cases, while five (12.5%) AHCR and three (7.5%) HHCR sockets overhung more than 12 mm. There was a minimal agreement between the two configurations (k = 0.28, *p* = 0.008) and a significant difference between the two configurations (*p* < 0.001) [Figure 2].

Anterior cup overhanging on the axial view in the middle of the superior half of the cup did not occur in 16 (40%) AHCR and 29 (72.5%) HHCR cases. Cups protruding 6–11 mm totaled 17 (42.5%) in the AHCR cohort and 6 (15%) in HHCR group, whereas only one (2.5%) ACHR socket protruded more than 12 mm. There was a moderate agreement between the two configurations (k = 0.32, *p* < 0.001) and a significant difference between the two configurations (*p* < 0.001) [Figure 3].

Anterior cup overhanging on the axial view at the center of the cup was not detected in 17 (42.5%) AHCR and 31 (77.5%) HHCR sockets. Prominence between 6 and 11 mm occurred in nine (22.5%) AHCR cups and one (2.5%) HHCR sockets. There was a minimal agreement between the two configurations (k = 0.23, *p* = 0.01) and a significant difference between the two configurations (*p* < 0.001) [Figure 4].

## 4. Discussion

Cup protrusion on at least one plane occurred frequently in high riding hip dysplasia, regardless the hip center of rotation. AHCR and HHCR cup configurations led to different patterns of cup prominence; sockets in AHCR had higher rates of protrusion on the sagittal view and on axial view, at three different sites. Moreover, the ACHR sockets had higher percentages of severe protrusion, in terms of magnitude.

The study has notable limitations. First, there is no assessment of the clinical relevance of cup overhanging: cup prominence is a necessary, but not a sufficient predisposing factor leading to psoas impingement [4,9]. Second, no anatomic description of the anterior acetabular wall was provided, thus the shape of the psoas valley (which is another notable causative factor of iliopsoas tendonitis) was not investigated [10,11]. Moreover, the CT evaluation of cup prominence was not based on fixed bony acetabular landmarks, as recommended by Brownlie et al., with the aim to reduce measurement mismatches to a minimum [12]. On the other hand, this is the first study providing data about cup overhanging in high riding dysplasia, with two cup configurations simulated in the same hip; this condition cannot be reproduced in clinical setting, and comparative clinical trials about dysplasia are impaired by the unique anatomic features of every hip. Similarly, the severe anatomic variations of high-grade dysplastic hips precluded the use of standard anatomic landmarks and the description of minute findings, such as the psoas valley. This drawback was partially obviated by providing multiple evaluation sites on the axial view, covering many of the possible positions of psoas impingement [10,11,12].

It is well-known that acetabular dysplasia, with the poor anterior–lateral bone stock and relative acetabular retroversion, may challenge cup position and may predispose one to anterior cup overhanging, however the influence of cup center height on anterior cup protrusion has not been established [10,11]. In this study, in every hip of a random pool of 40 Crowe III and IV hips, CT simulations of cup placement in anatomic and high hip center were performed. A low threshold for HHCR definition was adopted (20 mm) [6]. Cup positioning was calibrated, aiming to reduce anterior cup overhanging to a minimum, while respecting the posterior–medial bone stock and achieving an acceptable combined anteversion. After the simulations, HHCR cups resulted in being significantly less anteverted, less medialized and less caudal than AHCR sockets (with no significant differences in cup size). Medialized and buried cups are less likely to cause protrusion and psoas impingement; thus, potentially, AHCR sockets would have been less likely to overhang [3,4,10,11,12]. However, despite the more favorable cup position, AHCR cups were more frequently prominent on at least one plane (92.5% vs. 77.5%). Moreover, AHCR cups significantly protruded in the sagittal and axial planes (at every site), usually with more magnitude, than HHCR sockets. In four sites out of five, the two cup configurations had no or minimal agreement, demonstrating the striking difference between the two configurations in terms of cup overhanging. Thus, a significant difference in cup prominence was detected when were adopted two cup configurations, different in terms of hip center height. The high hip center, which provides superior–lateral bony coverage in DDH to support primary socket stability, was also effective in reducing the rates of anterior overhanging in comparison to anatomic center. This difference seems to overlook the single acetabular morphologies. However, the clinical impact of cup protrusion on iliopsoas impingement is still debated in DDH. While some authors noticed this association in a clinical setting (investigating selected cases of iliopsoas impingement after THA), Zhu et al. did not report an increased rate of iliopsoas tendonitis after THAs due to dysplasia [5,11]. In a comparison between THAs in dysplastic (Crowe II, III and IV) and non-dysplastic hips at a mid-term follow-up, the dysplastic cups, which were placed in anatomic or slightly elevated position, did not result in higher rates of iliopsoas impingement (2.6% versus 0.8%, respectively); however, the case series size was quite modest [5]. Thus, it can be concluded that, according to Dora et al. and Cyteval et al., cup protrusion is a necessary, but not sufficient predisposing factor in iliopsoas impingement after THA; in DDH, anatomic cup position may lead to an increased rate of cup overhanging and may consequentially raise the clinical risk for iliopsoas impingement. However, the choice of the center of rotation height should be carefully evaluated. A high hip center was associated with increased polyethylene wear, acetabular loosening, dislocation rate and inferior clinical outcomes [6,13]. Other authors achieved satisfying long-term outcomes in severe DDH using a slightly elevated center of rotation and appropriate offset reconstruction [6,14,15,16]. While the appropriate center of rotation height in DDH has not been established, there is no evidence for a routine adoption of high hip center, and the possible anecdotical risk of iliopsoas impingement is not sufficient to shift the current paradigm.

Iliopsoas tendonitis should be considered a clinical consequence of cup overhanging in specific acetabular morphologies. Thus, while choosing the appropriate cup placement in high degree DDH (namely considering cup stability, bone stock preservation, biomechanical reconstruction, offset restoration and wear reduction), anterior cup overhanging should be checked, especially when the center of rotation height is <20 mm. When anterior cup overhanging occurs, opportune measures to reduce the risk of iliopsoas impingement should be taken according to the specific cases, as asymmetrically curved cups, cup re-orientation or intra-operative iliopsoas release.

## Figures and Tables

**Figure 1 jcm-12-00606-f001:**
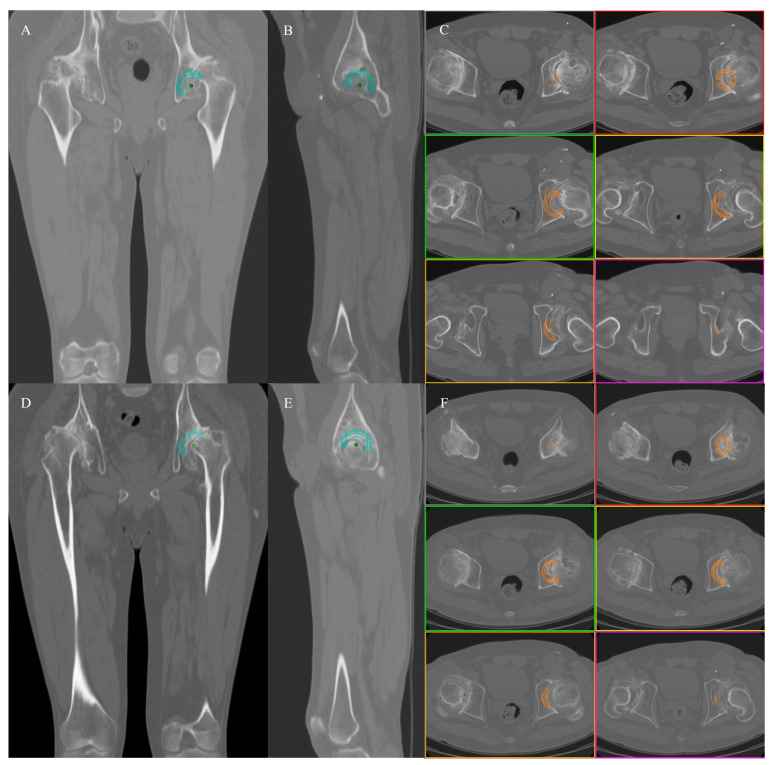
AHCR (**A**–**C**) and HHCR (**D**–**F**) cup planning in the same Crowe III patient: cup overhanging was assessed in the frontal view (**A**,**D**), in the sagittal view (**B**,**D**) and in the axial view (**C**,**F**).

**Figure 2 jcm-12-00606-f002:**
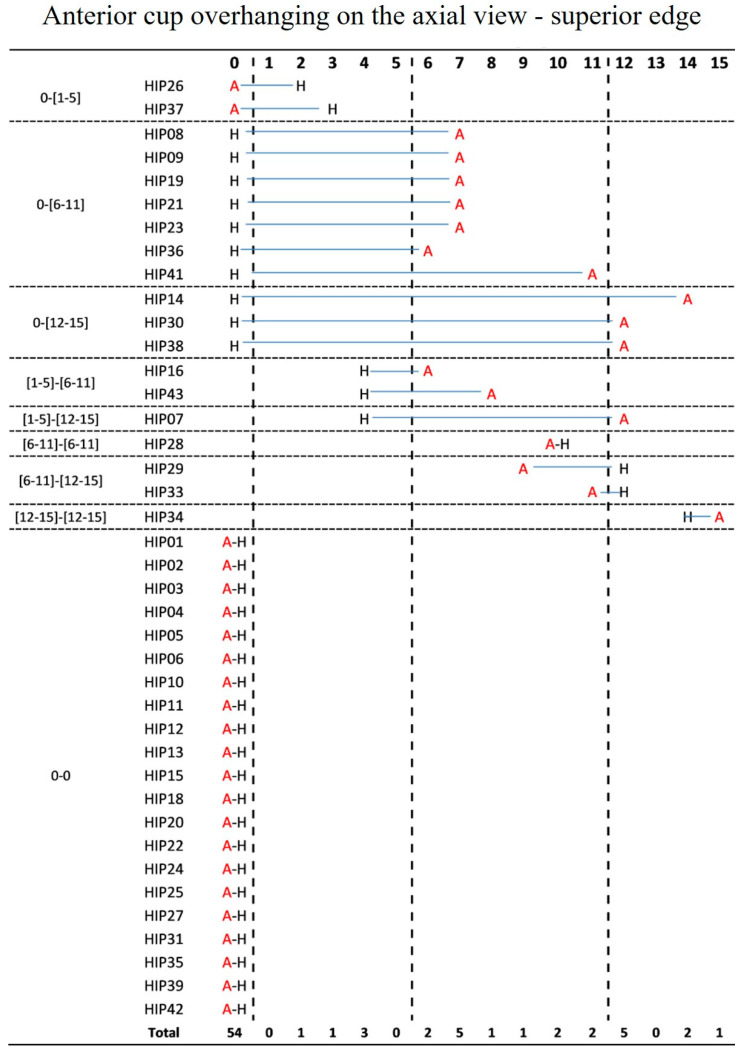
Anterior cup overhanging on the axial view at the superior edge of the cup in the same case according to the two configurations AHCR (A) and HHCR (H).

**Figure 3 jcm-12-00606-f003:**
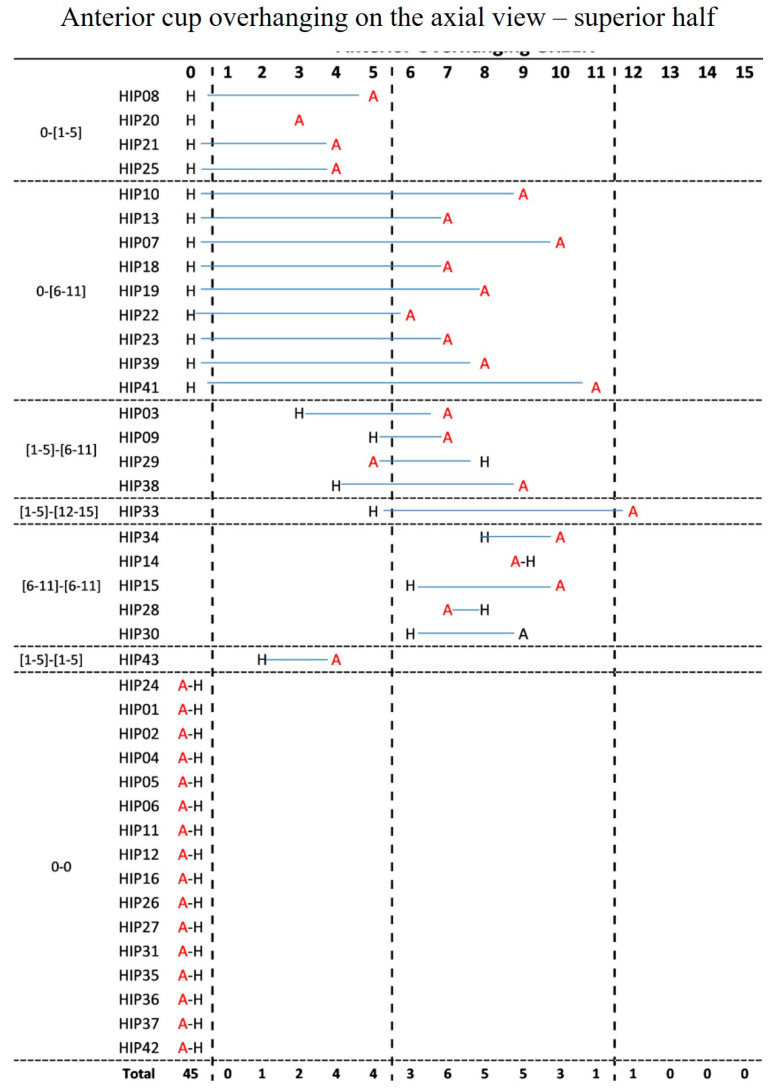
Anterior cup overhanging on the axial view in the middle of the superior half of the cup in the same case according to the two configurations AHCR (A) and HHCR (H).

**Figure 4 jcm-12-00606-f004:**
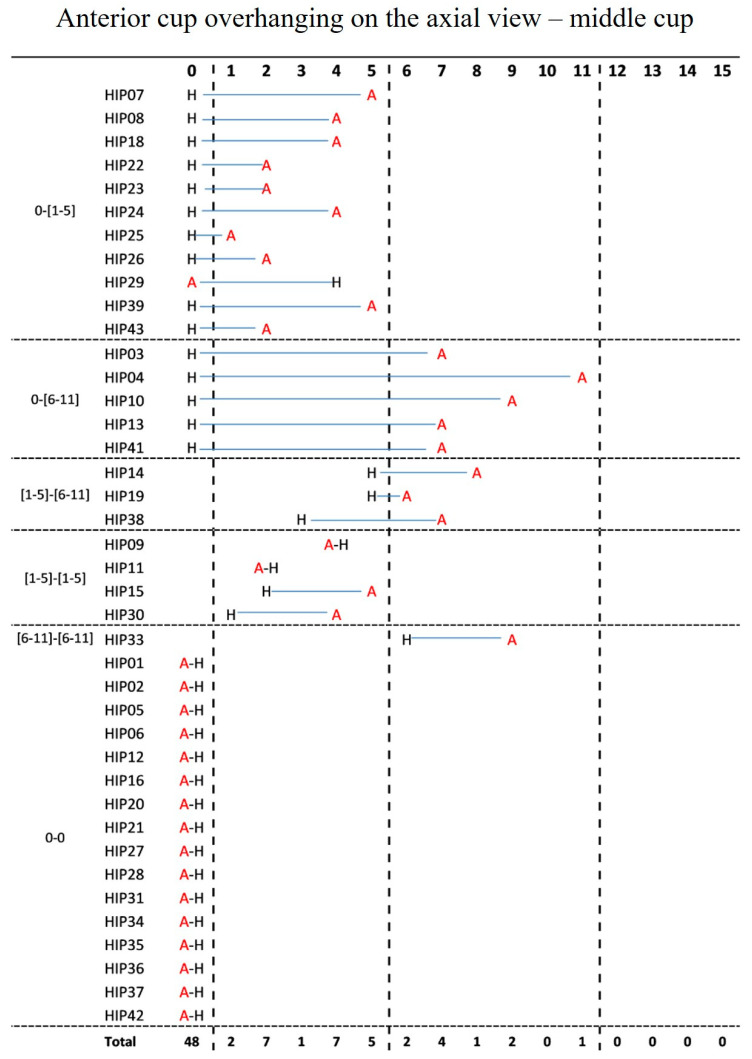
Anterior cup overhanging on the axial view at the center of the cup in the same case according to the two configurations AHCR (A) and HHCR (H).

**Table 1 jcm-12-00606-t001:** Cup prominence on every considered plane were stratified according to the entity and the AHCR and HHCR configurations.

	**Superior Overhanging on the Frontal View**				
	**HHCR**				
	**mm**	**0**	**1–5**	**6–11**	**12–15**
AHCR	0	12.5%	7.5%	12.5%	0
	1–5	0%	0%	5%	0
	6–11	15%	2.5%	20%	10%
	12–15	7.5%	2.5%	5%	0
	**Anterior Overhanging on the Sagittal View**				
	**HHCR**				
	**mm**	**0**	**1–5**	**6–11**	**12–15**
AHCR	0	40%	0	5%	0
	1–5	27.5%	2.5%	2.5%	0
	6–11	10%	5%	5%	0
	12–15	2.5%	0	0	0
	**Anterior Overhanging on the Axial View (Superior Edge of the Cup)**				
	**HHCR**				
	**Mm**	**0**	**1–5**	**6–11**	**12–15**
AHCR	0	52.5%	5%	0%	0
	1–5	0%	0	0%	0
	6–11	17.5%	5%	2.5%	5%
	12–15	7.5%	2.5%	0	2.5%
	**Anterior Overhanging on the Axial View (Middle of the Superior Half of the Cup)**				
	**HHCR**				
	**mm**	**0**	**1–5**	**6–11**	**12–15**
AHCR	0	40%	0	0	0
	1–5	10%	2.5%	2.5%	0
	6–11	22.5%	7.5%	12.5%	0
	12–15	0	2.5%	0	0
	**Anterior Overhanging on the Axial View (Center of the Cup)**				
	**HHCR**				
	**mm**	**0**	**1–5**	**6–11**	**12–15**
AHCR	0	40%	2.5%	0	0
	1–5	25%	10%	0	0
	6–11	12.5%	7.5%	2.5%	0
	12–15	0	0	0	0

## Data Availability

Data is contained within the article.

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
