# Peer review of "Cup Overhanging in Anatomic Socket Position or High Hip Center of Rotation in Total Hip Arthroplasty for Crowe III and IV Dysplasia: A CT-Based Simulation"

_jcm, 2023, doi:10.3390/jcm12020606_

Round 1
Reviewer 1 Report
Your work has been carefully evaluated. Your work about Cup overhanging in anatomic socket position or high hip cen- 2 ter of rotation in total hip arthroplasty for Crowe III and IV 3 dysplasia: a CT-based simulation. There are comments as follow
Strong point :. This paper is very Interesting and fascinating topic with good English. No need for native speaker to correct the English. If the author present some pictures that is relevant to the description for example: pictures of AHCR or HHCR would be spectacular. Also, the method was very well-planned and comprehensive.
Opportunity to improve : Some description might be hard to understand for some non- orthopedist to read and understand. Providing some background to some procedures might be helpful.
Author Response
Point 1. Strong point :. This paper is very Interesting and fascinating topic with good English. No need for native speaker to correct the English. If the author present some pictures that is relevant to the description for example: pictures of AHCR or HHCR would be spectacular. Also, the method was very well-planned and comprehensive.
Response 1. Thank you. An example was given in the first picture (Figure 1), when the same case was simulated in anatomical and high hip center position.
Point 2. Opportunity to improve : Some description might be hard to understand for some non- orthopedist to read and understand. Providing some background to some procedures might be helpful.
Response 2. We tried to rephrase the simulation techniques adopted for AHCR and HHCR, in order to make them more clear. The paper deals with special cases in THA and can be complex even for non-experienced orthopaedic surgeons.
The pae was modified: line 82: In AHCR, the acetabular component was placed in the true acetabulum with the aim to place the center of rotation in the anatomic position: the anatomic position was cross-checked by assessing the center of rotation height, which should be <20 mm, as previously stated in another paper [7]. Differently, in HHCR, the cup center of rotation was positioned outside the true acetabulum (and the anatomic center of rotation of the hip), where the superolateral bony coverage was most adequate: a center of rotation height ≥20 mm from the inter-teardrop line (vertical distance) was required [7].
Reviewer 2 Report
I have some questions for the authors:
1. In introduction, I think that the problem of iliopsoas impingement must be mentioned, I mean, if there are some papers on the matter, what percentage is the real problem. If there are some paper, then yes the problem should be studied.
I explain it because I have some experience in total hip after hip dyslocation, and in my opinion, I need to put the socket in center of rotacion, regardless of possible iliopsoas impingement.
2. Materials:
I would like to know if the patients with the dysplastic hip (III or IV), had already undergo surgery before starting the study. If so, I would like to know what percentage of groin pain they had.
I know that this aspect has not been studied, and this is pointed in the limitations, but I would like to know their opinion about it.
3. discussion:
They say: "...however the roe of cup center height has not been established (10,11)."
I think that this point is well studied in the literature. If you can place the cup in the center of rotation, the biomechanism of the hip will get improve.
I woluld like that authors point out the clinical applicability. I think this aspect very important.
Author Response
1. In introduction, I think that the problem of iliopsoas impingement must be mentioned, I mean, if there are some papers on the matter, what percentage is the real problem. If there are some paper, then yes the problem should be studied.
I explain it because I have some experience in total hip after hip dyslocation, and in my opinion, I need to put the socket in center of rotacion, regardless of possible iliopsoas impingement.
Response 1. The problem of ilipsoas impingement in THA after dysplastic hips is probably overlooked. Only one paper by Zhu studied the incidence, but the size of the cohort was quite small. However, many papers highlighted that dysplastic cases should be more prone to IP impingement, potentially. As you observed, in high riding dysplasia, the problem of IP impingement is far less important that other issues, like bone stock, primary component fixation, implant stability, stem choice, need of adjuctive procedures and so. In these cases, IP issues may be masked by other more relevant factors.
Introduction was modified: line 47: There is a paucity of literature providing data about the incidence of iliopsoas impingement and possible causative factors in THA for DDH: only Zhu et al. noticed that iliopsoas impingement occurred in 2.6% of the THAs performed in highly dysplastic hips with no prior surgical release, whereas the incidence in non-dysplastic hips dropped to 0.8% [5]. To date, there is no study investigating the incidence of cup overhanging in DDH when the cup was positioned reproducing the anatomic hip center of rotation (AHCR) or when a high hip center of rotation (HHCR) was adopted.
Point 2. Materials: I would like to know if the patients with the dysplastic hip (III or IV), had already undergo surgery before starting the study. If so, I would like to know what percentage of groin pain they had. I know that this aspect has not been studied, and this is pointed in the limitations, but I would like to know their opinion about it.
Response 2. 29 hips out of 40 underwent THA. In all of them, an high hip center was adopted. To our knowledge, none of them complained of groin pain. Cup overhanging in DDH was investigated after noticing that in some cases when anatomic hip center was adopted, the cup protrusion looked much more evident than in high hip center and the anterior bone stock provided less coverage.
Paper was not modified
Point 3: discussion: They say: "...however the roe of cup center height has not been established (10,11)." I think that this point is well studied in the literature. If you can place the cup in the center of rotation, the biomechanism of the hip will get improve. I woluld like that authors point out the clinical applicability. I think this aspect very important
Response 3. The sentence was confusing: we are referring to cup height and anterior cup prominence, not to cup center height in general. The sentence was rephrased
The paper was modified. Line 230: It is well-known that acetabular dysplasia, with the poor anterior-lateral bone stock and relative acetabular retroversion, may challenge cup position and may predispose to anterior cup overhanging, however the influence of cup center height on anterior cup protrusion has not been established
The clinical and practical aspects of the paper were stressed and the conclusion was partially rewritten.
Paper was modified. Line 253: While some Authors noticed this association in clinical setting (investigating selected cases of iliopsoas impingement after THA), Zhu et al. did not report increased rate of iliopsoas tendonitis after THAs due to dysplasia [5,11]. In a comparison between THAs in dysplastic (Crowe II, III and IV) and non-dysplastic hips at a mid-term follow-up, the dysplastic cups, which were placed in anatomic or slightly elevated position, did not result in higher rates of iliopsoas impingement (2.6% versus 0.8%, respectively): however, the case series size was quite modest [5]. Thus, it can be concluded that, according to Dora et al. and Cyteval et al., cup protrusion is a necessary, but not sufficient predisposing factor in iliopsoas impingement after THA: in DDH, anatomic cup position may lead to an increased rate of cup overhanging and may consequentially raise the clinical risk for iliopsoas impingement. However, iliopsoas tendonitis should be considered a clinical consequence of cup overhanging in specific acetabular morphologies. Thus, while choosing the appropriate cup placement in high degree DDH (namely considering cup stability, bone stock preservation, biomechanical reconstruction, offset restoration and wear reduction), anterior cup overhanging should be checked, especially when an anatomic position is adopted. When anterior cup overhanging occurs, opportune measures to reduce the risk of iliopsoas impingement should be taken according to the specific case, as asymmetrically curved cups, cup re-orientation or intra-operative iliopsoas release.